# Therapeutic Effect of Rapamycin on TDP-43-Related Pathogenesis in Ischemic Stroke

**DOI:** 10.3390/ijms24010676

**Published:** 2022-12-30

**Authors:** Yi-Syue Tsou, Jing-Huei Lai, Kai-Yun Chen, Cheng-Fu Chang, Chi-Chen Huang

**Affiliations:** 1Ph.D. Program in Medical Neuroscience, College of Medical Science and Technology, Taipei Medical University, Taipei 110, Taiwan; 2Department of Neurosurgery, Taipei Medical University Hospital, Taipei 110, Taiwan; 3Taipei Neuroscience Institute, Taipei Medical University, Taipei 110, Taiwan; 4Core Laboratory of Neuroscience, Office of R&D, Taipei Medical University, Taipei 110, Taiwan; 5Center for Neurotrauma and Neuroregeneration, Taipei Medical University, Taipei 110, Taiwan; 6Department of Neurosurgery, Taipei City Hospital, Zhong Xiao Branch, Taipei 110, Taiwan

**Keywords:** TDP-43, rapamycin, ischemic stroke

## Abstract

Stroke is a major cause of death and disability across the world, and its detrimental impact should not be underestimated. Therapies are available and effective for ischemic stroke (e.g., thrombolytic recanalization and mechanical thrombectomy); however, there are limitations to therapeutic interventions. Recanalization therapy has developed dramatically, while the use of adjunct neuroprotective agents as complementary therapies remains deficient. Pathological TAR DNA-binding protein (TDP-43) has been identified as a major component of insoluble aggregates in numerous neurodegenerative pathologies, including ALS, FTLD and Alzheimer’s disease. Here, we show that increased pathological TDP-43 fractions accompanied by impaired mitochondrial function and increased gliosis were observed in an ischemic stroke rat model, suggesting a pathological role of TDP-43 in ischemic stroke. In ischemic rats administered rapamycin, the insoluble TDP-43 fraction was significantly decreased in the ischemic cortex region, accompanied by a recovery of mitochondrial function, the attenuation of cellular apoptosis, a reduction in infarct areas and improvements in motor defects. Accordingly, our results suggest that rapamycin provides neuroprotective benefits not only by ameliorating pathological TDP-43 levels, but also by reversing mitochondrial function and attenuating cell apoptosis in ischemic stroke.

## 1. Introduction

Stroke is the second leading cause of death in the World Health Organization’s 2-2-global health estimate, leading to severe long-term disability and death worldwide, and incurring huge social costs. Stroke is therefore a serious health problem that requires further attention to attenuate death and disability. Among all strokes, approximately 87% are ischemic strokes, which are usually caused by blood clots that occlude the blood supply from the cerebral artery in the brain and initiate ischemic cascades [1]. Reperfusion after ischemia, which involves the instant recanalization of occluded vessels by mechanical or chemical therapies, is essential for preventing irreversible brain damage and has been proven to be the most effective therapy for stroke treatment. However, reperfusion after ischemic stroke may reverse the benefits of endovascular recanalization by exacerbating brain injury, resulting in the extension of brain damage, which is called cerebral ischemia/reperfusion (I/R) injury [2,3]. A high percentage of ischemic stroke patients die from extensive brain damage caused by reperfusion [4,5]. Improvements in recanalization techniques and the development of adjunct neuroprotective agents targeting antioxidation and anti-inflammation are indispensable to improve the clinical outcomes of ischemic patients. I/R injury induces a series of harmful consequences, including blood–brain barrier destruction, cerebral hemorrhage, neurovascular damage and neuronal death, which are the main causes of neurological deficits and mortality [4]. A wide spectrum of pathological factors is involved, including oxidative stress, leukocyte infiltration, inflammation and apoptosis [6]. The main cause of hemorrhagic transformation is the breakdown of vessel integrity distal to occlusion. The goal for eliminating stroke-related neuroinflammation responses is the development of an effective adjunct therapy to improve adverse results after I/R. In addition to neuroinflammation, among I/R injuries, the oxidative stress resulting from the overproduction of reactive oxygen species (ROS) in mitochondria is regarded as the primary event leading to excessive oxidative damage and neuronal death in cerebral I/R injury [7,8]. Hence, the imbalance between oxidants/antioxidants and mitochondrial dysfunction are fundamental causes of cerebral I/R injury. Thus, exploring potential therapeutic strategies targeting antioxidant, anti-inflammatory, and anti-apoptotic pathways will provide a novel approach to the treatment of ischemic stroke.

TDP-43 pathology is identified as the major component of the UBIs aggregated in numerous neurodegenerative diseases, especially in frontotemporal lobar degeneration (FTLD-U) and amyotrophic lateral sclerosis (ALS) [9,10]. TAR DNA-binding protein 43 (TDP-43) is a 43-kDa protein predominantly expressed in the nucleus and is a DNA/RNA binding factor involved in multiple cellular functions, including transcriptional repression, RNA splicing and translation [11,12]. Pathological TDP-43 protein is abnormally hyperphosphorylated, ubiquitinated, and cleaved into C-terminal fragments to form insoluble cytosolic aggregates in the central nervous system, including the hippocampus, frontal cortex, temporal cortex and spinal-cord motor neurons [10]. Numerous disease-related characteristics are observed in TDP-43 neuropathies, including mitochondrial dysfunction, endoplasmic reticulum (ER) stress, oxidative stress, gliosis and proteolysis dysfunction. TDP-43 aggregates show prion-like cell-to-cell propagation, which may lead to disease progression [13] The involvement of TDP-43 in neurodegenerative diseases is well established. However, little is known about the role of the pathological TDP-43 in ischemic stroke. Thus, there is an urgent need to explore novel potential therapeutic strategies to expand the neural-protective effects and improve the final outcomes for stroke treatment. 

It has been shown that the mammalian target of the rapamycin (mTOR) cascade is associated with I/R injury [14]. The mTOR plays an important role in regulating numerous cellular functions, including cell growth, proliferation, protein synthesis and autophagy activation [15]. Rapamycin, an antibiotic and immunosuppressant drug, is applied in the treatment of cardiovascular diseases and the prevention of allografts from rejection after organ transplantation. It plays a neuroprotective role by enhancing autophagy activities through mTOR inhibition to remove toxic protein aggregates, thereby restoring the process of neuronal degeneration and disease progression in Alzheimer’s disease, Parkinson’s disease and Huntington’s disease [16,17]. A growing body of studies has demonstrated that rapamycin protects neurons not only via neuronal autophagy, but also through other unknown effects. By enhancing autophagy activities, the accumulation of unwanted and toxic aggregates is attenuated and, consequently, this prevents the process of neuronal degeneration and brain disease progression. In TDP-43 proteinopathies, rapamycin reduces the accumulation of truncated TDP-43 and rescues TDP-43 mislocalization through the induction of autophagy. The phenotypes of motor defects are also improved after rapamycin treatment in these animal models [18]. These results strongly indicate that the activation of autophagy is a potential therapeutic strategy for TDP-43-related neuropathies. Based on the enhanced expression patterns of pathological insoluble TDP-43 observed in mouse brains with acute ischemic stroke, we were interested in further exploring the therapeutic efficacy of rapamycin for eliminating TDP-43-mediated pathology in ischemic stroke. In the middle cerebral artery occlusion rat model of ischemic stroke, the infarct areas were decreased and motor impairments were ameliorated after rapamycin treatment. Furthermore, we found that secondary injuries and motor deficits were significantly attenuated in ischemic rats administered with rapamycin. Further investigations suggested that, in addition to decreased pathological TDP-43, accompanied by increasing levels of BDNF, mitochondrial antioxidant enzyme superoxide dismutase 2 (SOD2) and mitochondria transcription factor A (mtTFA) were also observed after the rapamycin treatment. Moreover, we found that rapamycin also reverses cell apoptosis by decreasing active caspase 3 and ROS production. Our findings provide a novel molecular mechanism correlated with the effect of rapamycin in attenuating the pathological TDP-43-induced neuropathies in ischemic stroke, and provide a better understanding for the development of therapeutic strategies for ischemic stroke. 

## 2. Results

### 2.1. Pathological TDP-43 Aggregates Specifically Expressed in the Damaged Infarcted Right Ischemic Cortex Regions in a Time-Dependent Manner

The pathological hallmark of insoluble TDP-43 proteinopathy has been shown to be the major component of UBIs aggregated in numerous neurodegenerative diseases, especially in 45% frontotemporal lobar degeneration (FTLD-U) patients and 95% amyotrophic lateral sclerosis (ALS) patients [9,10]. However, the pathological role of TDP-43 in ischemic stroke remains unclear. To identify the patterns of pathological TDP-43 fraction after ischemia in an animal model, we determined the expression levels of the insoluble TDP-43 species in different brain regions of ischemic rats. Transient focal cerebral ischemia was induced by using the right MCA (middle cerebral artery) occlusion model in rats. After 15, 30, 60 and 90 min of ischemia, the brain tissues were dissected from these ischemic rats, and the soluble and insoluble fractions of TDP-43 were extracted by using sequential biochemical fractionation. We found that the insoluble full-length TDP-43 in the urea fraction was significantly increased in the right ischemic cortex region in a time-dependent manner, but not in the left non-ischemic cortex region (Figure 1). In addition, the induction of insoluble TDP-43 in the urea fraction was not observed in either the basolateral or hippocampus regions (Figure 2). We also analyzed the expression pattern of phosphorylated TDP-43 in the post-ischemic brain, since phosphorylated TDP-43 is an important diagnostic marker for neuropathology. Abundant evidence has proven that phosphorylated TDP-43 potentiates a number of neurotoxic effects, including changes in RNA splicing, cytoplasmic mislocalization and insoluble TDP-43 aggregate formation, leading to neurodegeneration. In addition, the phosphorylated TDP-43 levels in CSF or plasma are significantly higher in ALS patients than those in normal healthy controls, suggesting that p-TDP-43 is a potential diagnostic biomarker or indicator for disease progression in ALS patients [19,20,21]. Here, we found that phosphorylated TDP-43 is also significantly increased after ischemia in a time-dependent manner (Appendix A). Our results showed that not only insoluble TDP-43, but also p-TDP-43, were specifically increased in the post-ischemic brain, suggesting that pathological TDP-43, including p-TDP-43 and the insoluble TDP-43 fraction, plays an important role in neuronal damage formation during ischemic stroke. 

### 2.2. Augmented SOD2 and Reactive Astrogliosis after Acute Ischemic Stroke

During ischemia reperfusion, ROS release substantially contributes to cell damage and death via caspase3-mediated apoptotic signals. The mitochondrial antioxidant enzyme superoxide dismutase 2 (SOD2) is the first-line defense against mitochondrial ROS release and, subsequently, reduces superoxide radicals and infarction volumes after cerebral ischemia [22]. Thus, we also examined the SOD2 expression levels and found that the levels of SOD2 proteins were decreased in the infarcted right ischemic cortex regions after 60 min of ischemia, suggesting that the increase in ROS and infarcted areas observed in the right ischemic cortex regions were highly correlated with the reduced level of SOD2 (Figure 3). Another hallmark of the brain’s response to ischemic injury and neurodegeneration is the activation of glial cells. To evaluate whether astrogliosis is induced by these insoluble TDP-43 inclusions in the ischemia model, we further determined the expression levels of the glia marker, GFAP. Transient focal cerebral ischemia was induced using the right MCA (middle cerebral artery) occlusion model in the rats. The levels of GFAP proteins were significantly increased in line with the upregulation of the TDP-43 protein in the infarcted right ischemic cortex regions after 60 min of ischemia, suggesting that the reactive astrogliosis induced by ischemic stress is highly correlated with the accumulation of TDP-43 protein (Figure 3). Here, we show that the enhancement of the pathological TDP-43 inclusion observed in ischemic brains regions is accompanied by the induction of astrogliosis and attenuated antioxidant SOD2 levels in I/R injury.

### 2.3. Rapamycin Rescued Cell Viability and Decreased ROS Production after OGD Hypoxia

Rapamycin, a selective inhibitor of the mammalian target of rapamycin (mTOR), has been reported to be beneficial in TDP-43-related neurodegenerative diseases, such as FTLD-U. Since we observed the induction of the TDP-43 protein after ischemic stress (Figure 1), we further tested the therapeutic effect of rapamycin on the OGD (oxygen and glucose deprivation) hypoxia cell culture model to mimic ischemia injury in vitro. SH-SY5Y cells were treated with different doses of rapamycin (0, 1, 10 ng/mL) for 24 h after 4 h incubation under OGD hypoxia or normoxia. The cell viability of the SH-SY5Y cells was examined by the MTT assay. The cell survival rates were increased in a dose-dependent manner in the cells exposed to OGD hypoxia with rapamycin administration, while no significant change was observed in the cells under normoxia treated with different doses of rapamycin (Figure 4). In addition, we also examined the effect of rapamycin on mitochondrial function and discovered that, in the cells exposed to OGD hypoxia, the ROS production was decreased in a dose-dependent manner after rapamycin treatment, while no significant effect of rapamycin was observed in the cells under normoxia (Figure 5). These results demonstrated the therapeutic benefit of rapamycin in reversing the cell death induced by OGD hypoxia and rescuing mitochondrial function, leading to the attenuation of ROS production.

### 2.4. Rapamycin Decreased Infarction Volume and Improved Body Asymmetry

The therapeutic efficacy of rapamycin in eliminating TDP-43-mediated pathology after ischemic stroke was further examined. After 60 min of ischemia, rats were injected intraperitoneally with rapamycin (250 μg/kg) for 24 h and then killed for further analysis. The control animals were injected with PBS (vehicle) in parallel. We then further analyzed the infarction size in the ischemic brain regions of the rats, and found that the infarction size in the right ischemic cortex was significantly decreased after rapamycin treatment compared to the vehicle control group (Figure 6). In addition, we also tested the body asymmetry to monitor the motor neuron function, and found that the percentage of the recovery in body asymmetry was dramatically improved after rapamycin treatment compared to the vehicle control group (Figure 7). Furthermore, in the rats intraperitoneally injected with rapamycin, the apoptotic cells stained in the brain sections with tunnels were significantly decreased (Figure 8), suggesting a therapeutic effect of rapamycin on pathological TDP-43-induced cell apoptosis. Our results strongly suggested that rapamycin can attenuate secondary injury and motor deficits after ischemic stroke.

### 2.5. The Molecular Mechanism of Rapamycin in Attenuating Pathological TDP-43-Induced Neuronal Defects after Ischemic Stroke

To determine whether the improvement in ischemic stroke with rapamycin treatment was due to the reduction in neuronal apoptosis, we examined the expression levels of active caspase-3 by using Western blot analysis. The protein lysates were assayed by immunoblotting and collected from the SH-SY5Y cells treated with different doses of rapamycin (0, 1, 10 ng/mL) for 24 h after 4 h of OGD hypoxia or normoxia incubation. Caspase-3 is activated in apoptotic cells, and we found that the levels of caspase-3 proteins were decreased in the SH-SY5Y cells treated with rapamycin under OGD hypoxia, while no significant change was observed in the cells under normoxia administered with different doses of rapamycin (Figure 9). In addition to caspase-3, the mitochondria transcription factor A (mtTFA), which is required for mitochondrial DNA transcription and maintenance, is also involved in cellular apoptosis [23]. The deregulation of mtTFA leads to enhanced oxidative damage, and its reduction is observed in areas of brain lesions in numerous neurodegenerative diseases [24,25]. Here, we found a more significant change of the expression levels of mtTFA in a dose-dependent manner after rapamycin treatment in the cells exposed to OGD hypoxia. These results suggest that rapamycin treatment regulates caspase-3 and mtTFA signaling cascades to attenuate neuronal apoptosis in an ischemic stroke model. We then further examined the SOD2 expression levels and found that the levels of SOD2 proteins showed an increased trend in cells treated with rapamycin under both normoxia and OGD hypoxia (Figure 9), replicating the mitochondria protection effect of rapamycin (Figure 9). Moreover, we further analyzed the efficacy of rapamycin in attenuating pathological TDP-43 in an ischemic stroke animal model. Transient focal cerebral ischemia was induced using the right MCA (middle cerebral artery) occlusion model in the rats. After 60 min of ischemia, the rats were injected intraperitoneally with rapamycin (250 μg/kg) for 24 h and then killed for further analysis. The control animals were injected with PBS (vehicle) in parallel. The brain tissues were dissected from these ischemic rats and the soluble and insoluble fractions of TDP-43 were extracted by sequential biochemical fractionation. The insoluble TDP-43 was decreased in the ischemic cortex region administered with rapamycin treatment, compared to the vehicle control group (Figure 10). Our results indicate that rapamycin treatment in cell culture models or ischemic rat models could exert an anti-apoptotic and antioxidant effect on ischemic stroke correlated with pathological TDP-43 inclusion. 

## 3. Discussion

It is known that a high percentage of ischemic stroke patients die from extensive brain damage caused by reperfusion [4,5]. The outcome of reperfusion therapy after ischemic stroke is not always favorable. A common complication after reperfusion with alteplase (recombinant tissue plasminogen activator) or endovascular therapy (EVT) is hemorrhagic transformation (HT). When hemorrhagic transformation occurs, it increases morbidity and mortality. The incidence ranges from 3 to 40%, according to different definitions and studies. The occurrence of hemorrhagic transformation is due to the destruction of blood–brain barrier (BBB) after ischemic stroke and the extravasation of peripheral blood into the brain after reperfusion. The risk factors include reperfusion therapy (thrombolysis and thrombectomy), stroke severity, hyperglycemia, hypertension and age [26]. Adjunct neuroprotective agents are pharmacological brain-protective agents combined with reperfusion therapy; their aims are to reduce brain cell death and decrease patient disability and stroke mortality. Although many agents (including nimodipine, verapamil, nerinetide, uric acid, 3-N-butylphtalide, human urinary kallidinogenase, allogeneic adult mesenchymal bone-marrow cells, JTR-161(allogeneic stem-cell product), dimethyl fumarate and fingolimod) are undergoing testing in clinical trials, none have been proven to be effective and safe for use [27].

Revascularization and the limitation of secondary neuronal injury are the primary goals of advanced stroke management. IV thrombolysis and endovascular therapy are the two main forms of revascularization for selected patients. A limited number of acute ischemic stroke (AIS) patients are suitable for these treatments, largely owing to the narrow therapeutic window. Only 3.2–5.2% of all AIS patients are treated with IV-tPA within 3 h of suffering an acute ischemic stroke in the United States [28]. The extension of the IV-tPA window from 3 to 4.5 h increased the utilization of IV-tPA by up to 20% [29]. Endovascular therapy further expanded the time window for AIS treatment. Since 2015, multiple trials have demonstrated improvements in the overall outcomes of AIS patients with proximal middle cerebral artery or internal carotid artery occlusion treated with EVT. These studies expanded the time windows from 6 h [30,31,32,33,34] and 8 h [35] to 12 h [36]. These ongoing efforts include increasing numbers of eligible AIS patients receiving revascularization. Thus, there is an urgent need to explore novel potential therapeutic strategies to expand the neural-protective effects and improve the final outcomes for stroke treatment. 

It has been shown that the mammalian target of the rapamycin (mTOR) cascade is associated with I/R injury [14]. The mTOR plays an important role in regulating numerous cellular functions, including cell growth, proliferation, protein synthesis and autophagy activation [15]. Numerous studies have proved that the activation of autophagy is an alternative neuroprotective strategy for traumatic spinal cord injury or neurodegenerative diseases. After decades of considering autophagy as a cell death pathway, autophagy has recently been discovered to have a survival function through the clearing of protein aggregates and damaged cytoplasmic organelles in response to a variety of stress conditions [37]. Most recently, increasing evidence from the literature revealed that autophagy induction offered protection against neurodegeneration and traumatic brain or spinal cord injury [38,39]. Autophagy sequesters damaged and dysfunctional organelles/proteins and transports them to lysosomes for degradation/recycling. This process provides nutrients for injured neurons. Other groups also discovered that two drugs, including spermidine and chloroquine, would promote healthy aging though interfering with the autophagy pathway [40,41]. The mTOR inhibitor rapamycin is known to have a therapeutic effect on ischemia stroke [14], however, its underlying molecular mechanism remains elusive. Previous studies have indicated that administration of rapamycin reduces body temperature [42,43]. In fact, hypothermia (32–35°C) can alleviate the ischemia–reperfusion injury of different organs such as brain, heart, liver, kidney, etc., reduce the generation of ROS and avoid cell apoptosis [44]. However, whether rapamycin-induced hypothermia is beneficial for prolonging the lifespan of ischemia stroke patients is less discussed. Since we observed the recovery of mitochondrial function, the attenuation of cellular apoptosis, a reduction in infarct areas and improvements in motor defects in ischemic rats administered rapamycin, this implies that rapamycin-induced hypothermia may be one of the underlying neuroprotective mechanism for ischemic stroke.

The nuclear protein TDP-43, which forms detergent-insoluble cytosolic aggregates in the central nervous system, including the hippocampus, cortex and spinal cord motor neurons, was recently identified as a major component of the UBIs aggregated in numerous neurodegenerative diseases [9,10]. TDP-43 is involved in multiple cellular functions, including transcriptional repression, RNA splicing and translation [11,12]. Pathological TDP-43 protein is abnormally hyperphosphorylated, ubiquitinated and cleaved into C-terminal fragments to form insoluble cytosolic aggregates in the central nervous system, including the hippocampus, frontal cortex, temporal cortex and spinal cord motor neurons [10]. Numerous disease-related characteristics are observed in TDP-43 neuropathies, including mitochondrial dysfunction, endoplasmic reticulum (ER) stress, oxidative stress, gliosis and proteolysis dysfunction. TDP-43 aggregates show prion-like cell-to-cell propagation, which may lead to disease progression [13]. The pathological TDP-43 protein sequestered from nuclei into cytosol cannot be degraded normally and is prone to forming cytosolic insoluble aggregates, which leads to a plethora of deleterious effects, ranging from mitochondrial dysfunction to cellular toxicity [45,46,47,48,49]. The accumulation of insoluble and phosphorylated TDP-43 fractions is believed to eventually lead to neurite loss and, subsequently, to neuronal death [50]. Recently, it was identified that increased TDP-43 expression induced mitochondrial dysfunction by suppressing the activity of mitochondrial complex I and reduced mitochondrial ATP synthesis, including decreased mitochondrial membrane potential and the elevated production of reactive oxygen species (ROS) in both cellular and animal models of TDP-43 proteinopathy, as well as in FTLD and ALS patients’ brain tissues [51]. Information about the involvement of TDP-43 in ALS and FTLD is broadly discussed. However, little is known about the role of the pathological TDP-43 in ischemic stroke. So far, it has been identified that the full-length TDP-43, but not the 25-kDa C-terminal fragment in the core region in the urea fraction, increases after acute ischemic stroke, and the sequestered TDP-43 in cytosol was co-localized with ubiquitin and caspase-3 signals in immunoblotting staining [52]. This is similar to the pathological signatures in TDP-43-related neurodegenerative diseases, such as FTLD and ALS. However, cytosolic ubiquitinated TDP-43 inclusion was not observed in the core ischemic region [52,53], suggesting that the cytoplasmic redistribution of insoluble TDP-43 without the formation of TDP-43 inclusions is a hallmark of ischemic stroke and that TDP-43 pathology has a distinct molecular signature after stroke, compared to chronic neurodegenerative TDP-43 proteinopathies, such as FTLD and ALS. 

Our results showed a time-dependent increase in phosphorylated and insoluble TDP-43 fractions in the ischemic stroke brain region, but not in the cortex contralateral to the ischemic side. The longer ischemic time caused greater pathological TDP-43 accumulation, including insoluble and phosphorylated TDP-43, suggesting that pathological TDP-43 plays a role in ischemic stroke. In the right cortex of an MCA (middle cerebral artery) occlusion rat model, the insoluble full-length TDP-43 in the urea fraction was increased immediately in the right ischemic cortex region after 15 min of ischemia, showing that the acute rise in TDP-43 is synchronized with acute ischemic stroke. We discovered that rapamycin could rescue cell viability after OGD hypoxia and decrease the infarction volume after ischemia/reperfusion injury, prompting us to evaluate whether TDP-43-related pathology is involved in rapamycin’s therapeutic mechanism in ischemic stroke. It has been clearly demonstrated that pathological TDP-43 is significantly degraded through the autophagy pathway, and that the activation of autophagy accelerates the clearance rate for the insoluble TDP-43 fraction and attenuates the neuropathy caused by TDP-43 [18,54,55]. We found that the insoluble TDP-43 fraction in the ischemic cortex was attenuated after rapamycin treatment by Western blotting. Furthermore, we found that the rapamycin rescued the cell viability after OGD hypoxia and decreased the infarction volume after ischemia/reperfusion injury. The neuroprotective role of rapamycin in cell viability and its antioxidant and mitochondrial functions occur through the modulation of caspase-3, SOD-2 and BDNF expression after I/R injury in SH-SY5Y. These results suggest that the impact of increased pathological TDP-43 observed in ischemic stroke is involved in impaired mitochondria and neuronal cell death, which can be reversed by rapamycin treatment. Here, we provided an alternative therapeutic mechanism for rapamycin in pathological TDP-43-related neuropathology in ischemic stroke. 

It is well known that aging is the most critical, comorbid, and unmodifiable risk factor for stroke and it affects subsequent treatment responses. In addition, lifestyle including high sugar diets, alcohol and tobacco addiction or high fat diets as well as ageing, brain injury, oxidative stress and neuroinflammation all negatively influence the onset, severity and duration of neurodegenerative diseases. As suggested, choosing a variety of healthy dietary components, such as polyunsaturated fatty acids and the antioxidants curcumin, resveratrol, blueberry polyphenols, sulphoraphane and salvionic acid, as well as caloric restriction and physical activity, will help us counteract ageing and associated neurodegenerative diseases via increasing autophagy and neurogenesis in the adult brain. In our study, we did not consider these influencing factors such as age, eating habits and lifestyle on stroke mechanism, and these influencing factors will be very good directions for our further research in the future.

## 4. Materials and Methods

### 4.1. In Vitro Cell Culture Assay

After 4 h of OGD (oxygen–glucose deprivation)-hypoxia incubation, SH-SY5Y cells were treated with different doses of rapamycin (0, 1, 10 ng/mL) for 24 h, after which the cell viability, ROS production and protein expression levels were determined. The normoxia groups were also treated with different doses of rapamycin (0, 1, 10 ng/mL) for 24 h.

### 4.2. Cell Viabilty Assay and ROS Production

Cytosolic ROS production was measured using 2’,7’-dichlorofluorescien diacetate (DCFDA) as previously described [56], which is converted into fluorescent DCF by oxidation. Cells were seeded in 6-well plates (1 × 10^6^ cells per well) and allowed to adhere overnight. After cell incubation under normoxia or OGD conditions in the presence or absence of emodin for 4 h, cells were incubated with DCFDA for 30 min at 37 °C and washed with PBS. DCF fluorescence was determined using a multi-well fluorescence spectrophotometer (Varioskan Flash, Thermo Scientific) with 490 nm excitation and 525 nm emission filters.

### 4.3. In Vivo Animal Models

We used 15 adult male Sprague Dawley rats (250~300 g) and randomly assigned three rats to different time points of ischemic injury (0, 15, 30, 60, 90 min). In addition, we used a further 24 adult male SD rats for analyzing body asymmetry, infarct brain volume and TUNEL staining, 12 of which were designated as a vehicle, while a further 12 rats were administered with rapamycin. Animals were housed under a 12-h light/dark cycle and provided with food and water ad libitum. All animal procedures were conducted in accordance with the National Institutes of Health under the approval of the Animal Care and Use Committee of the Taipei Medical University, Taipei, Taiwan (protocol: LAC-2016-0495).

### 4.4. Middle Cerebral Artery (MCA) Ligation

Adult male Sprague Dawley rats were anesthetized with chloral hydrate (400 mg/kg, i.p., initially and 100 mg/kg). The right MCA was ligated with a 10-O suture, using methods previously described [57,58]. The ligature was removed after 60 min of ischemia to generate reperfusion injury. After 24 h of reperfusion, the brain tissues were collected for further analysis. Using this animal model, our group generated consistent ischemic damage in rodent brains, as demonstrated by brain infarction visualization and behavioral analysis [59,60].

### 4.5. Treatment

After 60 min of ischemia and reperfusion injury (60 min I/R injury), rats received intraperitoneal injections of rapamycin (250 μg/kg) for 24 h and were then killed for further analysis [61,62,63]. The control animals were injected with PBS (vehicle) in parallel.

### 4.6. Histological Evaluation of Stroke Damage

The size of infarction was evaluated by triphenyltetrazolium chloride (TTC) staining, as described by [64]. After rapamycin treatment for 24 h, following 60 min I/R injury, some animals were killed and perfused intracardially with saline. The brain tissues were then removed, immersed in cwee saline for 5 min, and sliced into 2.0 millimeter thick sections. The brain slices were incubated in 2% triphenyltetrazolium chloride (TTC, Sigma-Aldrich, St. Louis, MO, USA), dissolved in normal saline for 10 min at room temperature, and then transferred into a 5% formaldehyde solution for fixation. The area of infarction on each brain slice was measured double blind using a digital scanner and the Image Tools program (University of Texas Health Sciences Center, San Antonio). The total infarction volume in each animal was obtained from the product of average slice thickness (2 mm) and sum of the area of infarction in all brain slices. We also measured tissue damage by thionin staining, and the area of non-infarcted tissue in the ischemic hemisphere was subtracted from the volume of the contralateral hemisphere. This method was corrected for post-ischemic edema.

### 4.7. Body Asymmetry

I/R-injury rats with or without rapamycin treatment were analyzed using an elevated body swing test [60,65,66,67]. Twenty trials of each rat were examined for lateral movements/turning when their bodies were suspended 20 cm above the testing table by lifting their tails. The frequency of initial turning of head or upper body contralateral to the ischemic side was counted in 20 consecutive trials. The maximum impairment in body asymmetry in stroke animals was 20 contralateral turns/20 trials. In normal rats, the average body asymmetry was 10 contralateral turns/20 trials (i.e., the animals turned in each direction with equal frequency). 

### 4.8. Sequential Biochemical Fractionation and Western Blotting

For preparation of the RIPA-soluble and urea fraction [68], the brain tissues were first lysed in a RIPA buffer (50 mM Tris-HCl pH 8.0, 150 mM NaCl, 1% NP-40, 0.5% sodium deoxycholate, 0.1% sodium dodecyl sulfate) freshly supplemented with complete EDTA-free protease inhibitor cocktail and phosphatase inhibitors (10 mM NaF and 1 mM Na_3_VO_4_). Next, the lysates were centrifuged at 4 °C for 15 min at 13,800× *g*. The supernatants were the RIPA-soluble fraction. The RIPA-insoluble pellets were washed twice in the RIPA buffer, re-sonicated and re-centrifuged. The washed pellets were finally dissolved in the urea buffer as the urea fraction. The total protein concentration was determined according to the Micro BCA procedure (Pierce, Rockford, IL, USA), using bovine serum albumin as standard. Equal amounts of proteins from each sample were electrophoresed on 4–12% polyacrylamide gels (Novex, San Diego, CA, USA) under reducing conditions and then transferred by electrotransfer onto polyvinylidene-difluoride membranes. The membranes were incubated in blocking buffer (20 mM Tris (pH 7.5), 137 mM NaCI, and 0.1% Tween-20 (TBST) with 5% (*w*/*v*) nonfat dry milk) overnight at 4 °C. The membranes were then incubated with primary antibody in TBS-T containing 1% nonfat dry milk for 1 h at room temperature or overnight at 4 °C. After a series of washes with TBST, the membranes were incubated with horseradish peroxidase-conjugated secondary antibody (Millipore, Livingston, UK) for 1 h at room temperature. The immunoblots were visualized using the ECL chemiluminescence-detection system. The animals experiencing sham surgery were regarded as experimental controls for comparisons with other experimental groups. The following primary antibodies were used: rat polyclonal anti-GFAP (BD Biosciences, CA, USA), anti-TDP-43 antibodies (Proteintech, IL, USA), anti-phospho-Ser409/410-TDP-43 antibodies (Proteintech, IL, USA), anti-SOD2 (Novus Biologicals, Littleton, CO, USA), anti-mTFA (Novus Biologicals, Littleton, CO, USA), anti-BDNF (Abcam, Cambridge, UK), anti-active Cap3 (Abcam, Cambridge, UK) and β-actin (Cell Signaling Technology, MA, USA).

### 4.9. TUNEL Staining

Animals were killed and their brains removed. One side of the brain was placed in 4% paraformaldehyde after transferring from 20 and 30% sucrose in PBS overnight. Brains were then sectioned. The brain was sectioned at 10~15 μm thickness and rinsed with PBS (3 × 10 min). TUNEL staining was performed by using a Roche assay kit (In situ Cell Death Detection Kit; Roche Diagnostic GmbH, Mannheim,. Germany), according to the manufacturer’s protocol

### 4.10. Statistical Analysis

All the experiments were conducted at least in triplicate, and the results are expressed as the mean ± standard error of the mean (SEM). Statistical analyses were conducted using one-way analysis of variance (ANOVA) or Student’s *t*-test. 

## Figures and Tables

**Figure 1 ijms-24-00676-f001:**
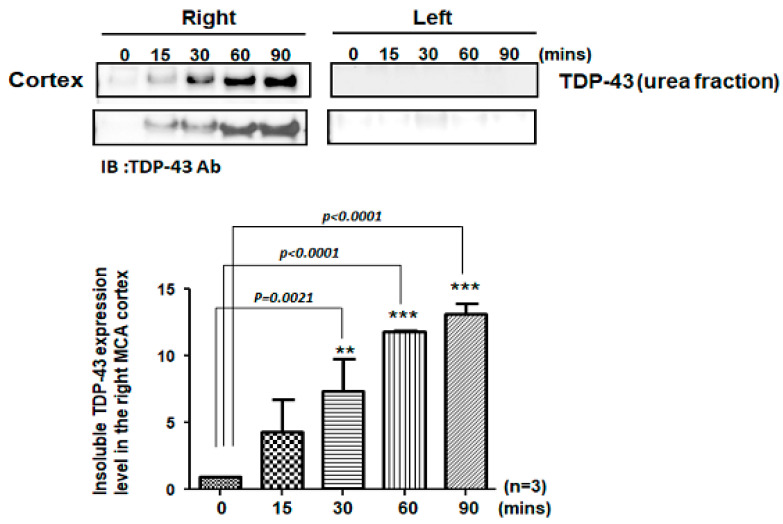
Pathological TDP-43 inclusion was specifically expressed in the right ischemic cortex region in a time-dependent manner. Transient focal cerebral ischemia was induced using the right MCA (middle cerebral artery) occlusion model in rats. After 15, 30, 60 and 90 min of ischemia, the brain tissues were dissected from these ischemic rats and the soluble and insoluble fractions of TDP-43 were extracted by sequential biochemical fractionation. The insoluble TDP-43 was significantly increased in the right ischemic cortex region in a time-dependent manner, but not in the left ischemic cortex region. Data are presented as the mean ± SEM (*n* = 3) (15, 30, 60 and 90 min compared to 0 min in the right ischemic cortex region by one-way ANOVA, ** *p* = 0.0021; *** *p* < 0.0001).

**Figure 2 ijms-24-00676-f002:**
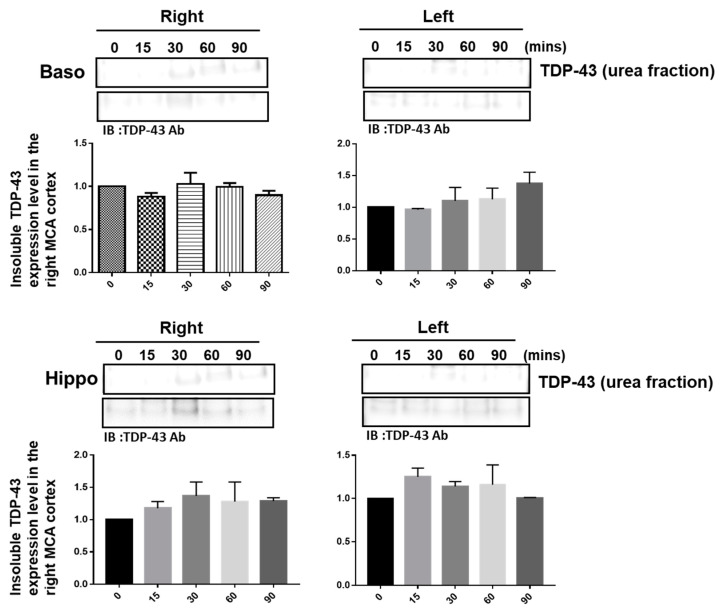
The induction of insoluble TDP-43 was not observed either in the right or in the left ischemic basolateral or hippocampus regions. Transient focal cerebral ischemia was induced using the right MCA (middle cerebral artery) occlusion model in rats. After 15, 30, 60 and 90 min of ischemia, the brain tissues were dissected from these ischemic rats and the soluble and insoluble fractions of TDP-43 were extracted by sequential biochemical fractionation. The expression pattern of insoluble TDP-43 was examined by specific antibodies. Data are presented as the mean ± SEM (*n* = 3) (15, 30, 60 and 90 min compared to 0 min in the basolateral and hippocampus regions by one-way ANOVA).

**Figure 3 ijms-24-00676-f003:**
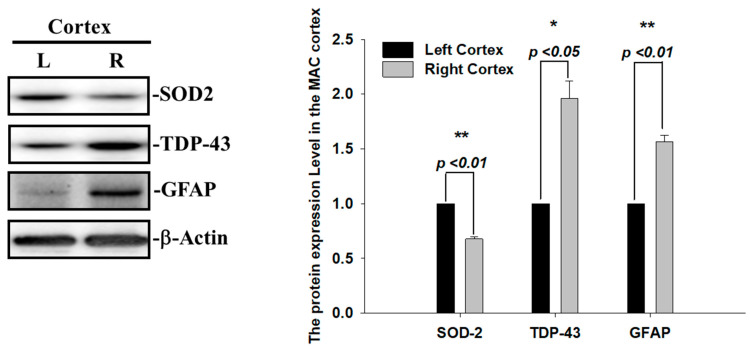
Augmented SOD2 and reactive astrogliosis after acute ischemic stroke. The expression patterns of SOD-2, GFAP and TDP-43 were examined by specific antibodies. β-actin was used as internal control. The levels of SOD-2 and GFAP protein were significantly increased in line with upregulation of TDP-43 protein in the infarcted right ischemic cortex regions after 60 min of ischemia. Data are presented as the mean ± SEM (*n* = 3) (right ischemic cortex region compared to left non-ischemic cortex region. These results were analyzed by two-tailed Student’s *t*-test, * *p* <0.05; ** *p* < 0.01).

**Figure 4 ijms-24-00676-f004:**
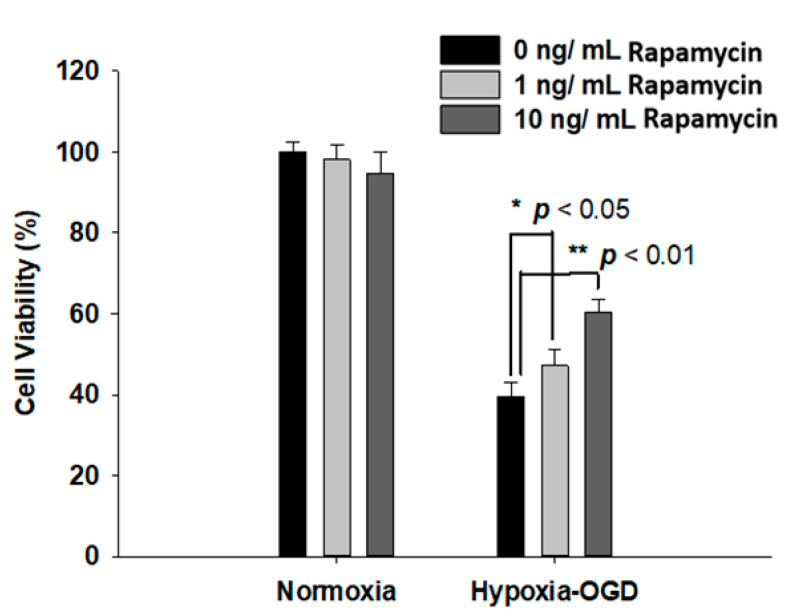
Rapamycin rescued the cell viability after OGD hypoxia. After 4 h of OGD hypoxia incubation, SH-SY5Y cells were treated with different doses of rapamycin (0, 1, 10 ng/mL) for 24 h, after which the cell viability was determined by MTT assay. The normoxia groups were also treated with different doses of rapamycin (0, 1, 10 ng/mL) for 24 h and the cell viability was determined by MTT assay. The cell viability of SH-SY5Y with rapamycin under OGD hypoxia incubation was the opposite of that shown by the normoxia group. Data are presented as the mean ± SEM (*n* = 4) (** *p* < 0.01; * *p* < 0.05, normoxia/OGD hypoxia with 1 or 10 ng/mL rapamycin compared to normoxia/OGD hypoxia without rapamycin). These results were analyzed by one-way ANOVA.

**Figure 5 ijms-24-00676-f005:**
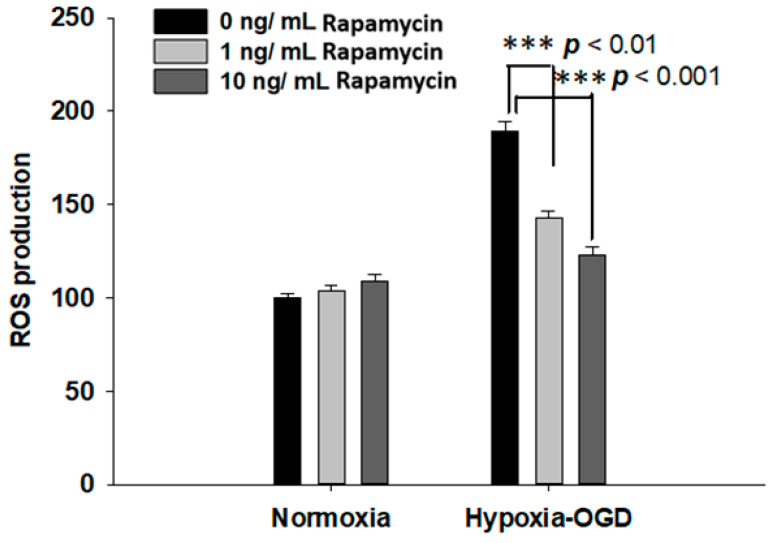
Rapamycin decreased the ROS production after OGD hypoxia. The ROS production was determined by DCDFA assay. The ROS of the SH-SY5Y cells were treated with different doses of rapamycin (0, 1, 10 ng/mL) for 24 h after 4 h of OGD hypoxia incubation and normoxia, after which the cell viability was determined by ROS assay. The ROS generation of SH-SY5Y with rapamycin under OGD hypoxia incubation was decreased compared to the normoxia group. Data are presented as the mean ± SEM (*n* = 4) (*** *p* < 0.01, normoxia/OGD hypoxia with 1 or 10 ng/mL rapamycin compared to normoxia/OGD hypoxia without rapamycin). These results were analyzed by one-way ANOVA.

**Figure 6 ijms-24-00676-f006:**
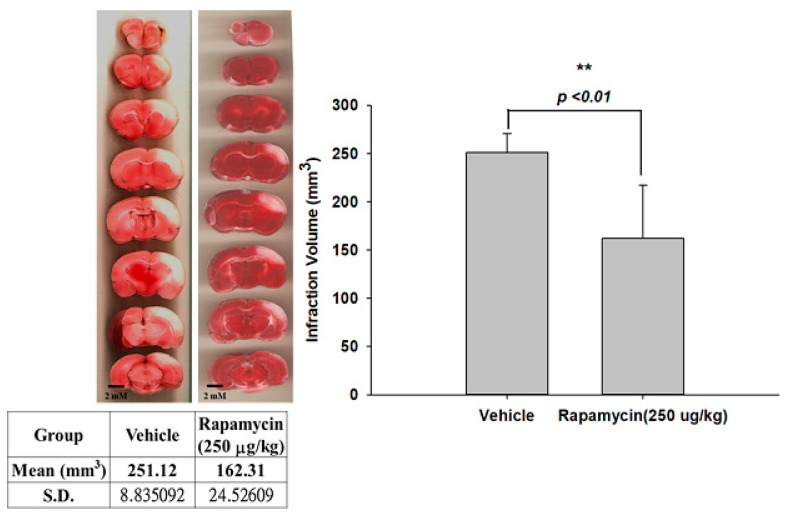
Rapamycin decreased the infarction volumes after I/R injury. After 60 min of ischemia, rats were injected intraperitoneally with rapamycin (250 μg/kg) for 24 h and then killed for further analysis. The control animals were injected with PBS (vehicle) in parallel. The infarction size in the right ischemic cortex was significantly decreased after rapamycin treatment compared to the vehicle control group. Data are presented as the mean ± SEM (*n* = 6, ** *p* < 0.01, rapamycin (250 μg/kg) compared to vehicle by two-tailed Student’s *t*-test).

**Figure 7 ijms-24-00676-f007:**
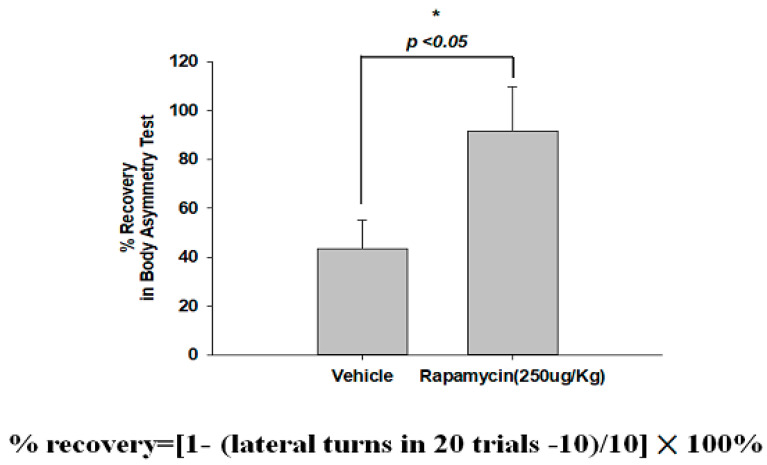
Rapamycin improved the body asymmetry after ischemia-reperfusion (I/R) injury. After 60 min of ischemia and reperfusion injury (60 min I/R injury), rats were administered an intraperitoneal injection of rapamycin (250 μg/kg) for 24 h and then killed for further analysis. The recovery in body asymmetry was dramatically improved after rapamycin treatment compared to the vehicle control group. Data represent means ± standard deviations of six independent experiments. * indicates *p* < 0.05 using Student’s *t* test. Data are presented as the mean ± SEM (*n* = 12, * *p* < 0.05, rapamycin (250 μg/kg) compared to vehicle by two-tailed Student’s *t*-test).

**Figure 8 ijms-24-00676-f008:**
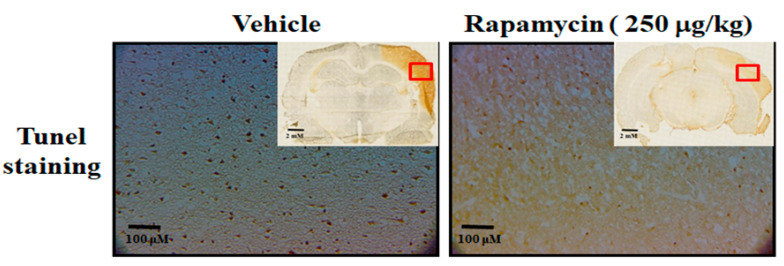
Rapamycin decreased apoptosis after ischemia–reperfusion (I/R) injury in the MCAo rodent model. After 60 min of ischemia and reperfusion injury (60 min I/R injury), rats were administered intraperitoneal injections of rapamycin (250 μg/kg) for 24 h and then killed for further analysis. The control animals were injected with PBS (vehicle) in parallel. The apoptotic cells were analyzed by tunnel staining within whole brains in the upper-right corner as indicated by red rectangle (*n* = 3).

**Figure 9 ijms-24-00676-f009:**
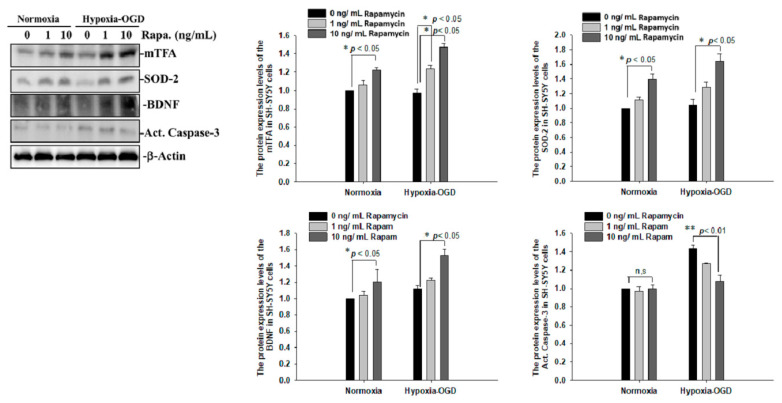
Rapamycin regulated the expression of mTFA, SOD-2, BDNF and Act. Caspase-3. The protein lysates were assayed by immunoblotting and collected from the SH-SY5Y cells treated with different doses of rapamycin (0, 1, 10 ng/mL) for 24 h after 4 h of OGD hypoxia incubation and normoxia. The levels of mTFA, SOD-2 and BDNF protein were significantly increased after the SH-SY5Y cells were treated with rapamycin. Caspase-3 proteins were decreased on the OGD hypoxia incubation. Data are presented as the mean ± SEM (*n* = 4) (** *p* < 0.01; * *p* < 0.05, normoxia/OGD hypoxia with 1 or 10 ng/mL rapamycin compared to normoxia/OGD hypoxia without rapamycin). These results were analyzed by one-way ANOVA.

**Figure 10 ijms-24-00676-f010:**
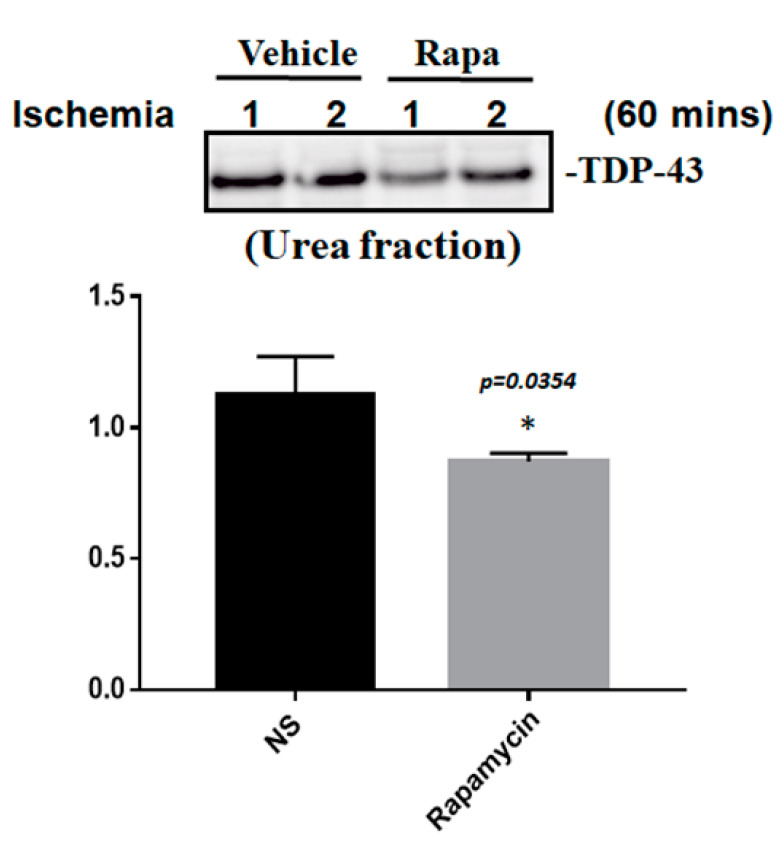
Rapamycin regulated the expression of pathological TDP-43. After 60 min of ischemia, rats were injected intraperitoneally with rapamycin (250 μg/kg) for 24 h and then killed for further analysis. The control animals were injected with PBS (vehicle) in parallel. The brain tissues were dissected from these ischemic rats and the soluble and insoluble fractions of TDP-43 were extracted by sequential biochemical fractionation. The insoluble TDP-43 was significantly decreased in the ischemic rats treated with rapamycin. Data are presented as the mean ± SEM (*n* = 3, * *p* = 0.0354, rapamycin (250 μg/kg) compared to the vehicle by two-tailed Student’s *t*-test).

## Data Availability

The datasets used and/or analyzed during the current study are available from the corresponding author on reasonable request.

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
