# Peer review of "Therapeutic Effect of Rapamycin on TDP-43-Related Pathogenesis in Ischemic Stroke"

_ijms, 2022, doi:10.3390/ijms24010676_

Round 1

Reviewer 1 Report

This is an interesting study on effects of rapamycin on TDP-43 mediated pathophysiology after focal stroke, induced by MCAO, in rats. While the data shows numerous parameters of in vitro and in vivo experiments to support the authors hypothesis, there are some significant flaws in the Results.

1. No mention is made of animal numbers in vivo, despite the claim that a power calculation was used to determine animal numbers in the Methods.

2. No measurements of body temperature was made and needs to be done in the in vivo rat studies. This is a critical variable and body temperature needs to be maintained both during and after stroke surgery as well as with drug measurement. There is extensive literature that rapamycin lowers core body temperature. Indeed, this is a major mechanism for rapamycin effects to extend longevity.

3. The ordinates (% recovery) in Fig 7 is not understandable. One uses numbers of paw placement turns where 20 trials in a normal animal gives 10 for each side and after MCAO, there is an asymmetry of up to 20 with no use of the paw contralateral to the MCAO lesion.

4. The paragraph on statistics (4.9) seems to be just taken from some other paper. Only “t tests” are noted in the figure legends, but the statistics paragraph cites use of ANOVA, linear regression and the Bonferroni correction. There is no citation for these tests in the Results. One should not use multiple “t tests” for more than 2 groups but use ANOVA instead.

5. Minor point – the English needs some correction as to use of definite vs indefinite articles and singular vs plural. It should be carefully proofread.

Author Response

Dear Reviewer 1:

Attached please find our revised manuscript Therapeutic effect of rapamycin on TDP-43-related pathogenesis in ischemic stroke “. It has been revised following the review comments.

The specifics of the revisions are detailed below:

Reviewer 1 Comments for the Author...

1). No mention is made of animal numbers in vivo, despite the claim that a power calculation was used to determine animal numbers in the Methods.

We apologize for our unclear explanation. Here, we used 15 adult male Sprague–Dawley rats (250~300 g) and randomly assigned three rats to different time points of ischemic injury (0, 15, 30, 60, 90 mins). In addition, we used a further 24 adult male SD rats for analyzing body asymmetry, infarct brain volume and TUNEL staining, 12 of which were designated as vehicle, while a further 12 rats were administered with rapamycin. We have addressed this part on p.13 in the “Materials and Methods” sections as indicated by red texts.

2). No measurements of body temperature was made and needs to be done in the in vivo rat studies. This is a critical variable and body temperature needs to be maintained both during and after stroke surgery as well as with drug measurement. There is extensive literature that rapamycin lowers core body temperature. Indeed, this is a major mechanism for rapamycin effects to extend longevity.

We appreciate the Reviewer’s constructive suggestions. We have to apologize for being unable to examine this question well due to limited revision time. However, we raise a few points of views to address the impact of rapamycin-induced low body temperature on ischemia stroke in Line 512-523 on p. 12-13 in the “Discussion” section. The mTOR inhibitor rapamycin is known to have therapeutic effect on ischemia stroke (Wu et al., 2018), however, its underlying molecular mechanism remains elusive. Previous literatures have indicated that administration of rapamycin reduces body temperature (Krakauer and Buckley, 2012; Tran et al., 2016). In fact, hypothermia (32-35°C) can alleviate the ischemia-reperfusion injury of different organs such as brain, heart, liver, kidney, etc., reduce the generation of ROS and avoid cell apoptosis (Omileke et al., 2021). However, whether rapamycin–induced hypothermia is beneficial for prolonging the lifespan of ischemia stroke patients is less discussed. Since we observed the recovery of mitochondrial function, the attenuation of cellular apoptosis, a reduction in infarct areas and improvements in motor defects in ischemic rats administered rapamycin, implicating that rapamycin–induced hypothermia may be one of the underlying neuroprotective mechanism for ischemic stroke.

3). The ordinates (% recovery) in Fig 7 is not understandable. One uses numbers of paw placement turns where 20 trials in a normal animal gives 10 for each side and after MCAO, there is an asymmetry of up to 20 with no use of the paw contralateral to the MCAO lesion.

We apologize for our unclear explanation. I/R-injury rats with or without rapamycin treatment were analyzed using an elevated-body-swing test (Borlongan et al., 1995; Borlongan et al., 1998; Chang et al., 2000; Luo et al., 2009). Twenty trials of each rat were examined for lateral movements/turning when their bodies were suspended 20 cm above the testing table by lifting their tails. The frequency of initial turning of head or upper body contralateral to the ischemic side was counted in 20 consecutive trials. The maximum impairment in body asymmetry in stroke animals was 20 contralateral turns/20 trials. In normal rats, the average body asymmetry was 10 contralateral turns/20 trials (i.e., the animals turned in each direction with equal frequency). We have addressed this part on p.14 in “Materials and Methods” sections as indicated by red texts.

4). The paragraph on statistics (4.9) seems to be just taken from some other paper. Only “t tests” are noted in the figure legends, but the statistics paragraph cites use of ANOVA, linear regression and the Bonferroni correction. There is no citation for these tests in the Results. One should not use multiple “t tests” for more than 2 groups but use ANOVA instead.

We appreciate the Reviewer’s kind comments and apologize for our unclear explanation and imprecise statistics. We have re-calculated and redone statistical analysis of our results. Also, the p values and the statistical analysis methods have been shown and addressed in our figures and figure legends. We have thoroughly re-edited the” Materials and Methods” and “Figure legends”, and upload our modified Figures with the revised manuscript and revised cover letter. 

5). Minor point – the English needs some correction as to use of definite vs indefinite articles and singular vs plural. It should be carefully proofread.

We appreciate the Reviewer’s kind comments. As suggested by the reviewer, the manuscript has already been thoroughly edited by the paid editing services at https://www.mdpi.com/authors/english before re-submission. .

References

Borlongan, C.V., D.W. Cahill, and P.R. Sanberg. 1995. Locomotor and passive avoidance deficits following occlusion of the middle cerebral artery. Physiology & behavior. 58:909-917.

Borlongan, C.V., H. Hida, and H. Nishino. 1998. Early assessment of motor dysfunctions aids in successful occlusion of the middle cerebral artery. Neuroreport. 9:3615-3621.

Chang, C.F., K.C. Niu, B.J. Hoffer, Y. Wang, and C.V. Borlongan. 2000. Hyperbaric oxygen therapy for treatment of postischemic stroke in adult rats. Experimental neurology. 166:298-306.

Krakauer, T., and M. Buckley. 2012. Intranasal rapamycin rescues mice from staphylococcal enterotoxin B-induced shock. Toxins. 4:718-728.

Luo, Y., C.C. Kuo, H. Shen, J. Chou, N.H. Greig, B.J. Hoffer, and Y. Wang. 2009. Delayed treatment with a p53 inhibitor enhances recovery in stroke brain. Annals of neurology. 65:520-530.

Omileke, D., D. Pepperall, S.W. Bothwell, N. Mackovski, S. Azarpeykan, D.J. Beard, K. Coupland, A. Patabendige, and N.J. Spratt. 2021. Ultra-Short Duration Hypothermia Prevents Intracranial Pressure Elevation Following Ischaemic Stroke in Rats. Frontiers in neurology. 12:684353.

Tran, C.M., S. Mukherjee, L. Ye, D.W. Frederick, M. Kissig, J.G. Davis, D.W. Lamming, P. Seale, and J.A. Baur. 2016. Rapamycin Blocks Induction of the Thermogenic Program in White Adipose Tissue. Diabetes. 65:927-941.

Wu, M., H. Zhang, J. Kai, F. Zhu, J. Dong, Z. Xu, M. Wong, and L.H. Zeng. 2018. Rapamycin prevents cerebral stroke by modulating apoptosis and autophagy in penumbra in rats. Annals of clinical and translational neurology. 5:138-146.

Hope you will find the manuscript now acceptable for publication in International Journal of Molecular Sciences. Thanks again for your tremendous editing efforts.

My best regards,

Sincerely,

Chi-Chen Huang

Reviewer 2 Report

The authors present a study that demonstrates the increase in TDP-43 in a post-ischaemic brain and demonstrate rapamycin as a potential neuroprotective therapy. 

See below my comments on the manuscript.

General

Overall, I am not convinced of the impact of the findings of the current study. While TDP-43 is known to increase in levels in the post-ischaemic brain, the current study did not demonstrate an increase of phosphorylated TDP-43 in the post-ischaemic brain. 

Notwithstanding, the current study fails to demonstrate if in fact elevated levels of TDP-43 are detrimental to acute stroke recovery in rats or if elevated TDP-43 is a naturally occurring phenomenon after acute infarction. 

Even though the authors present data that they claim rapamycin as a potential neuroprotective agent through the reduction in TDP-43 post-ischaemia, I am not convinced with the conclusions offered by the authors as they have failed to consider known neuroprotective mechanisms of rapamycin. As it currently stands, rapamycin is neuroprotective, not because it acts via the direct reduction of TDP-43 but rather through complementing pathways. 

Specific major 

Line 16: Statement is misleading and directly contradicts information offered in the Introduction. Stroke therapies are available and effective for ischaemic stroke (e.g., thrombolyic recanalisation and mechanical thrombectomy), however, there are limitations to therapeutic interventions.

 Line 18: Not sure what the aurhtors are referring to with respect to adjunct neuroprotective agents. Please be specific

Line 27: Neuroprotective mechanism of rapamycin is not via the reduction of TDP43 rather through alternative pathways. 

Line 49: Would be of reader benefit to provide statistics of hemorrhagic transformation eluded to. 

Statements are misleading as they do not provide the full picture. Please be more descriptive with respect to statistics and controls

Line 60: Statement is incorrect. Main cause of haemorrhagic transformation is breakdown of vessel integrity distal to occlusion

Line 88:The Authors have failed to provide a logical exaplaination as to how the chronic accumulation of TDP43 is associated with worsening acute stroke 

Line 92: therapeutic window is 4.5 hours

Line 94: References utilised to describe trends in ischaemic stroke care are significantly outdated. on average they are 14 years old and a significant paradigm shift in the management of acute ischaemic stroke care has occurred in the meantime. The absence of updated stroke management pipelines and critical care statistics is a significant disservice to this manuscript. 

Line 106:  Neuronal autophagy is not a neuroprotective pathway or cascade

Line 109: The statement that activation of neuronal autophagy is factually incorrect and questions the Authors understanding of molecular pathways that contribute to separate neuroprotection or cell death.

Overall results: Sections are repetative of both introduction and methods. 

No statistical measures presented in any results section (e.g., fold reduction/increase in presented blots)

 Line 260: What was the rational to adminisiter rapamycin IP as opposed to IV?

Figure 8: What brain region was Figure 8 taken from?

Figure legend states micrographs were taken from MCAO rodents, however, the rounding of the window in the figures implies these are from a tissue culture well. 

Methods general: Methods section requires significant editing to ensure experimental procedures can be completed by independent laboratories. As it currently stands, all methods are superficially presented. 

Author Response

Dear Reviewer 2:

Attached please find our revised manuscript Therapeutic effect of rapamycin on TDP-43-related pathogenesis in ischemic stroke “. It has been revised following the review comments.

The specifics of the revisions are detailed below:

Reviewer 2 Comments for the Author...

Overall, I am not convinced of the impact of the findings of the current study. While TDP-43 is known to increase in levels in the post-ischaemic brain, the current study did not demonstrate an increase of phosphorylated TDP-43 in the post-ischaemic brain. 

We apologize for this unclear explanation. Yes, as suggested by the reviewer, we should analyze the expression pattern of phosphorylated TDP-43 in the post-ischemic brain, since phosphorylated TDP-43 is an important diagnostic marker for neuropathology. Abundant evidence has proven that phosphorylated TDP-43 potentiates a number of neurotoxic effects, including changes in RNA splicing, cytoplasmic mislocalization and insoluble TDP-43 aggregate formation, leading to neurodegeneration. In addition, the phosphorylated TDP-43 levels in CSF or plasma are significantly higher in ALS patients than those in normal healthy controls, suggesting that p-TDP-43 is a potential diagnostic biomarker or indicator for disease progression in ALS patients (Goossens et al., 2015; Hasegawa et al., 2008; Ren et al., 2021). Here, we found that phosphorylated TDP-43 is also significantly increased after ischemia in a time-dependent manner (Fig. S1). Our results showed that not only insoluble TDP-43, but also p-TDP-43, were specifically increased in the post-ischemic brain, suggesting that pathological TDP-43, including p-TDP-43 and insoluble TDP-43 fraction, plays an important role in neuronal-damage formation during ischemic stroke. Hope our findings can provide more convincing evidence for the vital effect of TDP-43 in ischemic stroke. We have addressed this part on p. 4 in the “Result” sections as indicated by red texts.

Notwithstanding, the current study fails to demonstrate if in fact elevated levels of TDP-43 are detrimental to acute stroke recovery in rats or if elevated TDP-43 is a naturally occurring phenomenon after acute infarction. 

We appreciate the Reviewer’s kind suggestions. The significant increase of insoluble full-length TDP-43 in urea fraction was observed in right ischemia cortex region only after 15 mins of ischemia, but not in 0 mins of ischemia right cortex or left non-ischemia cortex, basolateral or hippocampus regions, suggesting that the accumulation of TDP-43 is a detrimental factor for acute stroke recovery. Under normal condition, TDP-43 protein is majorly expressed as RIPA soluble form within nucleus, but not observed in urea fraction. However, under pathological condition, the hallmark of insoluble TDP‑43 aggregates that can be detected in the CSF and extracted by urea buffer only is significantly increased in numerous neurodegenerative diseases, including ALS, FTLD and Alzheimer’s diseases. Here we observed that insoluble TDP-43 in urea fraction is strictly accumulated in ischemia brain regions only, suggesting that TDP-43 is not a naturally occurring phenomenon after acute infarction, but is detrimental to ischemia stroke.

Even though the authors present data that they claim rapamycin as a potential neuroprotective agent through the reduction in TDP-43 post-ischaemia, I am not convinced with the conclusions offered by the authors as they have failed to consider known neuroprotective mechanisms of rapamycin. As it currently stands, rapamycin is neuroprotective, not because it acts via the direct reduction of TDP-43 but rather through complementing pathways. 

We apologize for this unclear explanation. We have answered this question in the following specific major comments.

Specific major 

Line 16: Statement is misleading and directly contradicts information offered in the Introduction. Stroke therapies are available and effective for ischaemic stroke (e.g., thrombolyic recanalisation and mechanical thrombectomy, however, there are limitations to therapeutic interventions.

We appreciate the Reviewer’s kind suggestions. As suggested by the reviewer, we have modified this sentence “So far, the effective therapies for ischemic stroke are still unavailable” in Line16 to “Therapies are available and effective for ischemic stroke (e.g., thrombolytic recanalization and mechanical thrombectomy); however, there are limitations to therapeutic interventions”.  

Line 18: Not sure what the aurhtors are referring to with respect to adjunct neuroprotective agents. Please be specific.

We apologize for this unclear explanation. Adjunct neuroprotective agents are pharmacological brain-protective agents combined with reperfusion therapy; their aims are to reduce brain-cell death and decrease patient disability and stroke mortality. Although many agents (including Nimodipine, Verapamil, Nerinetide, Uric Acid, 3-N-butylphtalide, Human Urinary Kallidinogenase, Allogeneic adult mesenchymal bone-marrow cells, JTR-161(allogeneic stem-cell product), Dimethyl Fumarate and Fingolimod) are undergoing testing in clinical trials, none have been proven to be effective and safe for use (Amado et al., 2022). We added a more-detailed description of this part in Line 58.-64. on p. 2 in "Introduction” section as indicated by red texts.

Line 27: Neuroprotective mechanism of rapamycin is not via the reduction of TDP-43 rather through alternative pathways. 

We apologize for this unclear explanation.

It has been clearly demonstrated that pathological TDP-43 is significantly degraded through the autophagy pathway, and that the activation of autophagy accelerates the clearance rate for insoluble TDP-43 fraction and attenuates the neuropathy caused by TDP-43 (Caccamo et al., 2009; Scotter et al., 2014; Wang et al., 2012). Actually, previous studies have implicated that the autophagy activator, rapamycin, is a potential therapeutic strategy for the disease progression of neurodegenerative diseases with TDP-43 proteinopathies via ameliorating insoluble TDP-43 aggregates and rescuing the pathological TDP-43-induced motor defect through induction of autophagy in vitro cell culture model and in vivo FTLD (Frontotemporal lobar demendia)-TDP-43 animal model (Wang et al., 2012). Here, we show that increased pathological TDP-43 fractions accompanied by impaired mitochondrial function and increased gliosis were observed in an ischemic-stroke rat model, suggesting a pathological role of TDP-43 in ischemic stroke. In ischemic rats administered rapamycin, the insoluble TDP-43 fraction was significantly decreased in the ischemic cortex region, accompanied by a recovery of mitochondrial function, the attenuation of cellular apoptosis, a reduction in infarct areas and improvements in motor defects. Accordingly, our results suggest that rapamycin provides neuroprotective benefits not only by ameliorating pathological TDP-43 levels, but also by reversing mitochondrial function and attenuating cell apoptosis in ischemic stroke.  We added a paragraph to discuss this subject on p.1 in “Abstract” and on p. 13 in the "Discussion” sections as indicated by red texts.

Line 49: Would be of reader benefit to provide statistics of hemorrhagic transformation eluded to. Statements are misleading as they do not provide the full picture. Please be more descriptive with respect to statistics and controls

We apologize for this unclear description. The outcome of reperfusion therapy after ischemic stroke is not always favorable. A common complication after reperfusion with alteplase (recombinant tissue plasminogen activator) or endovascular therapy (EVT) is hemorrhagic transformation (HT). When hemorrhagic transformation occurs, it increases morbidity and mortality. The incidence ranges from 3 to 40%,according to different definitions and studies. The occurrence of hemorrhagic transformation is due to the destruction of blood–brain barrier (BBB) after ischemic stroke and the extravasation of peripheral blood into the brain after reperfusion. The risk factors include reperfusion therapy (thrombolysis and thrombectomy), stroke severity, hyperglycemia, hypertension, and age (Spronk et al., 2021). We added a more-detailed description of this part in Line 49-58 on p.2 in the "Introduction” section as indicated by red texts.

Line 60: Statement is incorrect. Main cause of haemorrhagic transformation is breakdown of vessel integrity distal to occlusion

We apologize for this wrong description. As suggested by the reviewer, we have modified this sentence “Ischemia stroke usually enhances strong neuroinflammatory response in which antigen-independent activation of T cells is the main cause to I/R injury (Zrzavy et al., 2018).” to “The main cause of haemorrhagic transformation is breakdown of vessel integrity distal to occlusion” in Line 72-73 on p.2.

Line 88: The Authors have failed to provide a logical exaplaination as to how the chronic accumulation of TDP43 is associated with worsening acute stroke

We apologize for this unclear explanation. In the right cortex of an MCA (middle cerebral artery) occlusion rat model, the insoluble full-length TDP-43 in the urea fraction was increased immediately in the right ischemic cortex region after 15 mins of ischemia, showing that the acute rise in TDP-43 is synchronized with acute ischemic stroke. We added a more-detailed description of this part in Linen 100-104 on p. 3 in the “Introduction” section as indicated by red texts.

Line 92: therapeutic window is 4.5 hours

We appreciate the Reviewer’s kind corrections. We added a more-detailed description of this part in Line 105-119 on p. 3 in the "Introduction” section as indicated by red texts.

Line 94: References utilised to describe trends in ischaemic stroke care are significantly outdated. on average they are 14 years old and a significant paradigm shift in the management of acute ischaemic stroke care has occurred in the meantime. The absence of updated stroke management pipelines and critical care statistics is a significant disservice to this manuscript. 

We apologize for this unclear explanation. Revascularization and the limitation of secondary neuronal injury are the primary goals of advanced-stroke management. IV thrombolysis and endovascular therapy are the two main forms of revascularization for selected patients. A limited number of acute ischemic stroke (AIS) patients are suitable for these treatments, largely owing to the narrow therapeutic window. Only 3.2–5.2% of all AIS patients are treated with IV-tPA within 3 hours of suffering an acute ischemic stroke in the United States (Del Zoppo et al., 2009). The extension of the IV-tPA window from 3 to 4.5 hours increased the utilization of IV-tPA by up to 20% (Lyerly et al., 2013). Endovascular therapy further expanded the time window for AIS treatment. Since 2015, multiple trials have demonstrated improvements in the overall outcomes of AIS patients with proximal middle cerebral artery or internal carotid artery occlusion treated with EVT. These studies expanded the time windows from 6 hours (Berkhemer et al., 2015; Bracard et al., 2016; Campbell et al., 2015; Mocco et al., 2016; Saver et al., 2015) and 8 hours (Jovin et al., 2015) to 12 hours (Goyal et al., 2015). These ongoing efforts include increasing numbers of eligible AIS patients receiving revascularization. We added a more-detailed description of this part in Line 105-119 on p. 3 in the "Introduction” section as indicated by red texts.

Line 106:  Neuronal autophagy is not a neuroprotective pathway or cascade

We apologize for the unclear explanation. Numerous studies have proved that the activation of autophagy is an alternative neuroprotective strategy for traumatic spinal cord injury or neurodegenerative diseases. After decades of considering autophagy as a cell-death pathway, autophagy has recently been discovered to have a survival function through the clearing of protein aggregates and damaged cytoplasmic organelles in response to a variety of stress conditions (Radad et al., 2015). Most recently, increasing evidence from the literature revealed that autophagy induction offered protection against neurodegeneration and traumatic brain or spinal-cord injury (Meng et al., 2013; Ray, 2020). Autophagy sequesters damaged and dysfunctional organelles/proteins, and transported them to lysosomes for degradation/recycling. This process provides nutrients for injured neurons. We added a more-detailed description of this part in Line 125-135 on p. 3 in the "Introduction” section as indicated by red texts.

Line 109: The statement that activation of neuronal autophagy is factually incorrect and questions the Authors understanding of molecular pathways that contribute to separate neuroprotection or cell death.

We appreciate the Reviewer’s kind suggestions. We have addressed this issue in above questions.

Overall results: Sections are repetative of both introduction and methods. 

We appreciate the Reviewer’s kind comments. As suggested by the reviewer, we have re-edited “Introduction” and " Methods " sections. 

No statistical measures presented in any results section (e.g., fold reduction/increase in presented blots).

We appreciate the Reviewer’s kind comments and apologize for our imprecise statistics and unclear explanations. We have re-calculated and redone the statistical analysis of all of our results, and the p values have been shown in our figures. We also thoroughly re-edited the ”Material and Methods” and “Figure legends”, and upload our modified Figure images with the revised manuscript and revised cover letter. 

Line 260: What was the rational to adminisiter rapamycin IP as opposed to IV?

We apologize for the unclear explanation. Our approach is to refer to these papers (Al Shoyaib et al., 2019; Ohshima et al., 2015; Woodrum et al., 2010) and has cited these references in the ”Material and Methods”.

Figure 8: What brain region was Figure 8 taken from? Figure legend states micrographs were taken from MCAO rodents, however, the rounding of the window in the figures implies these are from a tissue culture well. 

We apologize for this ambiguity. We used the tunnel staining to determine the apoptosis in ischemic stroke rats. We showed the brain region from the ipsilateral cortex region and added the whole brain on the upper right corner.

Methods general: Methods section requires significant editing to ensure experimental procedures can be completed by independent laboratories. As it currently stands, all methods are superficially presented. 

We appreciate the Reviewer’s kind comments and apologize for the unclear explanation. As suggested by the reviewer, this part was addressed in greater detail and we thoroughly re-edited the " Material and Methods " sections. 

Hope you will find the manuscript now acceptable for publication in International Journal of Molecular Sciences. Thanks again for your tremendous editing efforts.

My best regards,

Sincerely,

Chi-Chen Huang

Reference

  1. Al Shoyaib, A., S.R. Archie, and V.T. Karamyan. 2019. Intraperitoneal Route of Drug Administration: Should it Be Used in Experimental Animal Studies? Pharmaceutical research. 37:12.
  2. Amado, B., L. Melo, R. Pinto, A. Lobo, P. Barros, and J.R. Gomes. 2022. Ischemic Stroke, Lessons from the Past towards Effective Preclinical Models. Biomedicines. 10.
  3. Berkhemer, O.A., P.S. Fransen, D. Beumer, L.A. van den Berg, H.F. Lingsma, A.J. Yoo, W.J. Schonewille, J.A. Vos, P.J. Nederkoorn, M.J. Wermer, M.A. van Walderveen, J. Staals, J. Hofmeijer, J.A. van Oostayen, G.J. Lycklama à Nijeholt, J. Boiten, P.A. Brouwer, B.J. Emmer, S.F. de Bruijn, L.C. van Dijk, L.J. Kappelle, R.H. Lo, E.J. van Dijk, J. de Vries, P.L. de Kort, W.J. van Rooij, J.S. van den Berg, B.A. van Hasselt, L.A. Aerden, R.J. Dallinga, M.C. Visser, J.C. Bot, P.C. Vroomen, O. Eshghi, T.H. Schreuder, R.J. Heijboer, K. Keizer, A.V. Tielbeek, H.M. den Hertog, D.G. Gerrits, R.M. van den Berg-Vos, G.B. Karas, E.W. Steyerberg, H.Z. Flach, H.A. Marquering, M.E. Sprengers, S.F. Jenniskens, L.F. Beenen, R. van den Berg, P.J. Koudstaal, W.H. van Zwam, Y.B. Roos, A. van der Lugt, R.J. van Oostenbrugge, C.B. Majoie, and D.W. Dippel. 2015. A randomized trial of intraarterial treatment for acute ischemic stroke. The New England journal of medicine. 372:11-20.
  4. Borlongan, C.V., D.W. Cahill, and P.R. Sanberg. 1995. Locomotor and passive avoidance deficits following occlusion of the middle cerebral artery. Physiology & behavior. 58:909-917.
  5. Borlongan, C.V., H. Hida, and H. Nishino. 1998. Early assessment of motor dysfunctions aids in successful occlusion of the middle cerebral artery. Neuroreport. 9:3615-3621.
  6. Bracard, S., X. Ducrocq, J.L. Mas, M. Soudant, C. Oppenheim, T. Moulin, and F. Guillemin. 2016. Mechanical thrombectomy after intravenous alteplase versus alteplase alone after stroke (THRACE): a randomised controlled trial. The Lancet. Neurology. 15:1138-1147.
  7. Caccamo, A., S. Majumder, J.J. Deng, Y. Bai, F.B. Thornton, and S. Oddo. 2009. Rapamycin rescues TDP-43 mislocalization and the associated low molecular mass neurofilament instability. The Journal of biological chemistry. 284:27416-27424.
  8. Campbell, B.C., P.J. Mitchell, T.J. Kleinig, H.M. Dewey, L. Churilov, N. Yassi, B. Yan, R.J. Dowling, M.W. Parsons, T.J. Oxley, T.Y. Wu, M. Brooks, M.A. Simpson, F. Miteff, C.R. Levi, M. Krause, T.J. Harrington, K.C. Faulder, B.S. Steinfort, M. Priglinger, T. Ang, R. Scroop, P.A. Barber, B. McGuinness, T. Wijeratne, T.G. Phan, W. Chong, R.V. Chandra, C.F. Bladin, M. Badve, H. Rice, L. de Villiers, H. Ma, P.M. Desmond, G.A. Donnan, and S.M. Davis. 2015. Endovascular therapy for ischemic stroke with perfusion-imaging selection. The New England journal of medicine. 372:1009-1018.
  9. Chang, C.F., K.C. Niu, B.J. Hoffer, Y. Wang, and C.V. Borlongan. 2000. Hyperbaric oxygen therapy for treatment of postischemic stroke in adult rats. Experimental neurology. 166:298-306.
  10. Del Zoppo, G.J., J.L. Saver, E.C. Jauch, and H.P. Adams, Jr. 2009. Expansion of the time window for treatment of acute ischemic stroke with intravenous tissue plasminogen activator: a science advisory from the American Heart Association/American Stroke Association. Stroke. 40:2945-2948.
  11. Goossens, J., E. Vanmechelen, J.Q. Trojanowski, V.M. Lee, C. Van Broeckhoven, J. van der Zee, and S. Engelborghs. 2015. TDP-43 as a possible biomarker for frontotemporal lobar degeneration: a systematic review of existing antibodies. Acta neuropathologica communications. 3:15.
  12. Goyal, M., A.M. Demchuk, B.K. Menon, M. Eesa, J.L. Rempel, J. Thornton, D. Roy, T.G. Jovin, R.A. Willinsky, B.L. Sapkota, D. Dowlatshahi, D.F. Frei, N.R. Kamal, W.J. Montanera, A.Y. Poppe, K.J. Ryckborst, F.L. Silver, A. Shuaib, D. Tampieri, D. Williams, O.Y. Bang, B.W. Baxter, P.A. Burns, H. Choe, J.H. Heo, C.A. Holmstedt, B. Jankowitz, M. Kelly, G. Linares, J.L. Mandzia, J. Shankar, S.I. Sohn, R.H. Swartz, P.A. Barber, S.B. Coutts, E.E. Smith, W.F. Morrish, A. Weill, S. Subramaniam, A.P. Mitha, J.H. Wong, M.W. Lowerison, T.T. Sajobi, and M.D. Hill. 2015. Randomized assessment of rapid endovascular treatment of ischemic stroke. The New England journal of medicine. 372:1019-1030.
  13. Hasegawa, M., T. Arai, T. Nonaka, F. Kametani, M. Yoshida, Y. Hashizume, T.G. Beach, E. Buratti, F. Baralle, M. Morita, I. Nakano, T. Oda, K. Tsuchiya, and H. Akiyama. 2008. Phosphorylated TDP-43 in frontotemporal lobar degeneration and amyotrophic lateral sclerosis. Annals of neurology. 64:60-70.
  14. Jovin, T.G., A. Chamorro, E. Cobo, M.A. de Miquel, C.A. Molina, A. Rovira, L. San Román, J. Serena, S. Abilleira, M. Ribó, M. Millán, X. Urra, P. Cardona, E. López-Cancio, A. Tomasello, C. Castaño, J. Blasco, L. Aja, L. Dorado, H. Quesada, M. Rubiera, M. Hernandez-Pérez, M. Goyal, A.M. Demchuk, R. von Kummer, M. Gallofré, and A. Dávalos. 2015. Thrombectomy within 8 hours after symptom onset in ischemic stroke. The New England journal of medicine. 372:2296-2306.
  15. Luo, Y., C.C. Kuo, H. Shen, J. Chou, N.H. Greig, B.J. Hoffer, and Y. Wang. 2009. Delayed treatment with a p53 inhibitor enhances recovery in stroke brain. Annals of neurology. 65:520-530.
  16. Lyerly, M.J., K.C. Albright, A.K. Boehme, R.B. Shahripour, J.T. Houston, P.V. Rawal, N. Kapoor, M. Alvi, A. Sisson, A.W. Alexandrov, and A.V. Alexandrov. 2013. The Potential Impact of Maintaining a 3-Hour IV Thrombolysis Window: How Many More Patients can we Safely Treat? Journal of neurological disorders & stroke. 1:1015.
  17. Meng, Y., Y. Yong, G. Yang, H. Ding, Z. Fan, Y. Tang, J. Luo, and Z.J. Ke. 2013. Autophagy alleviates neurodegeneration caused by mild impairment of oxidative metabolism. Journal of neurochemistry. 126:805-818.
  18. Mocco, J., O.O. Zaidat, R. von Kummer, A.J. Yoo, R. Gupta, D. Lopes, D. Frei, H. Shownkeen, R. Budzik, Z.A. Ajani, A. Grossman, D. Altschul, C. McDougall, L. Blake, B.F. Fitzsimmons, D. Yavagal, J. Terry, J. Farkas, S.K. Lee, B. Baxter, M. Wiesmann, M. Knauth, D. Heck, S. Hussain, D. Chiu, M.J. Alexander, T. Malisch, J. Kirmani, L. Miskolczi, and P. Khatri. 2016. Aspiration Thrombectomy After Intravenous Alteplase Versus Intravenous Alteplase Alone. Stroke. 47:2331-2338.
  19. Ohshima, M., A. Taguchi, H. Tsuda, Y. Sato, K. Yamahara, M. Harada-Shiba, M. Miyazato, T. Ikeda, H. Iida, and M. Tsuji. 2015. Intraperitoneal and intravenous deliveries are not comparable in terms of drug efficacy and cell distribution in neonatal mice with hypoxia-ischemia. Brain & development. 37:376-386.
  20. Radad, K., R. Moldzio, M. Al-Shraim, B. Kranner, C. Krewenka, and W.D. Rausch. 2015. Recent advances in autophagy-based neuroprotection. Expert review of neurotherapeutics. 15:195-205.
  21. Ray, S.K. 2020. Modulation of autophagy for neuroprotection and functional recovery in traumatic spinal cord injury. Neural regeneration research. 15:1601-1612.
  22. Ren, Y., S. Li, S. Chen, X. Sun, F. Yang, H. Wang, M. Li, F. Cui, and X. Huang. 2021. TDP-43 and Phosphorylated TDP-43 Levels in Paired Plasma and CSF Samples in Amyotrophic Lateral Sclerosis. Frontiers in neurology. 12:663637.
  23. Saver, J.L., M. Goyal, A. Bonafe, H.C. Diener, E.I. Levy, V.M. Pereira, G.W. Albers, C. Cognard, D.J. Cohen, W. Hacke, O. Jansen, T.G. Jovin, H.P. Mattle, R.G. Nogueira, A.H. Siddiqui, D.R. Yavagal, B.W. Baxter, T.G. Devlin, D.K. Lopes, V.K. Reddy, R. du Mesnil de Rochemont, O.C. Singer, and R. Jahan. 2015. Stent-retriever thrombectomy after intravenous t-PA vs. t-PA alone in stroke. The New England journal of medicine. 372:2285-2295.
  24. Scotter, E.L., C. Vance, A.L. Nishimura, Y.B. Lee, H.J. Chen, H. Urwin, V. Sardone, J.C. Mitchell, B. Rogelj, D.C. Rubinsztein, and C.E. Shaw. 2014. Differential roles of the ubiquitin proteasome system and autophagy in the clearance of soluble and aggregated TDP-43 species. Journal of cell science. 127:1263-1278.
  25. Spronk, E., G. Sykes, S. Falcione, D. Munsterman, T. Joy, J. Kamtchum-Tatuene, and G.C. Jickling. 2021. Hemorrhagic Transformation in Ischemic Stroke and the Role of Inflammation. Frontiers in neurology. 12:661955.
  26. Wang, I.F., B.S. Guo, Y.C. Liu, C.C. Wu, C.H. Yang, K.J. Tsai, and C.K. Shen. 2012. Autophagy activators rescue and alleviate pathogenesis of a mouse model with proteinopathies of the TAR DNA-binding protein 43. Proc Natl Acad Sci U S A. 109:15024-15029.
  27. Woodrum, C., A. Nobil, and S.L. Dabora. 2010. Comparison of three rapamycin dosing schedules in A/J Tsc2+/- mice and improved survival with angiogenesis inhibitor or asparaginase treatment in mice with subcutaneous tuberous sclerosis related tumors. Journal of translational medicine. 8:14.
  28. Zrzavy, T., J. Machado-Santos, S. Christine, C. Baumgartner, H.L. Weiner, O. Butovsky, and H. Lassmann. 2018. Dominant role of microglial and macrophage innate immune responses in human ischemic infarcts. Brain pathology (Zurich, Switzerland). 28:791-805.

Round 2

Reviewer 1 Report

Although the study would be strengthened if body temperature was monitored and measured, there is no way to do this after completion of these experiments. The remainder of my concerns are well addressed.

Author Response

Dear Reviewer 1:

We appreciate your constructive suggestions to make our manuscript better and more complete. Our revised manuscript has undergone English language editing by MDPI before the first time revision. Please find our English editing certificate in the attachment. Hope you will find the manuscript now acceptable for publication in International Journal of Molecular Sciences. Thanks again for your tremendous editing efforts.

My best regards,

Sincerely,

Chi-Chen Huang
